# The True Nature of the Energy Calibration for Nuclear Resonant Vibrational Spectroscopy: A Time-Based Conversion

**Hongxin Wang** [1,*,†] , **Yoshitaka Yoda** [2,†] **and Jessie Wang** [3,†]

1 SETI Institute, Mountain View, CA 94043, USA
2 Research and Utilization Division, SPring-8/JASRI, 1-1-1 Kouto, Sayo, Hyogo 679-5198, Japan
3 School of Computer Science, Georgia Institute of Technology, Atlanta, GA 30332, USA
* Correspondence: hongxin.ucd@gmail.com
† These authors contributed equally to this work.

**Abstract:** Nuclear resonant vibrational spectroscopy (NRVS) is an excellent synchrotron-based vibrational spectroscopy. Its isotope specificity and other advantages are particularly good to study, for example, iron center(s) inside complicated molecules such as enzymes. In order to investigate some small energy shifts, the energy scale variation from scan to scan must be corrected via an in-situ measurement or with other internal reference peak(s) inside the spectra to be calibrated. On the other hand, the energy re-distribution within each scan also needs attention for a sectional scan which has a different scanning time per point in different sections and is often used to measure weak NRVS signals. In this publication, we: (1) evaluated the point-to-point energy re-distribution within each NRVS scan or within an averaged scan with a time-scaled (not energy-scaled) function; (2) discussed the errorbar contributed from the improper "distribution" of $\Delta E_i$ or the averaged $\Delta E$ within one scan ($E_{err}1$) vs. that due to the different $\Delta E_i$ from different scans ($E_{err}2$). It is well illustrated that the former ($E_{err}1$) is as important as, or sometimes even more important than, the latter ($E_{err}2$); and (3) provided a procedure to re-calibrate the published NRVS-derived PVDOS spectra in case of need. This article establishes the concept that, at least for sectional NRVS scans, the energy positions should be corrected according to the time scanned rather than be scaled with a universal constant, as in a conventional calibration procedure.

**Keywords:** nuclear resonant vibrational spectroscopy; NRVS; time-based energy correction; energy calibration; in-situ energy calibration; vibrational zero energy position; $\Delta E$; energy scale





## 1. Introduction

Calibrating energies is a critical matter for all spectroscopies, including nuclear resonant vibrational spectroscopy (NRVS), which is synchrotron radiation (SR)-based modern vibrational spectroscopy with a lot of distinguished advantages [1–9]. In this publication, we provided a brief review on the energy calibration issues for NRVS, discussed the point-by-point energy calibration within one NRVS scan with different data acquisition time per point in different sections, and established the concept that (at least for the sectional scans) the energies should be calibrated according to the time scanned rather than scaled with a universal constant.

### 1.1. Nuclear Resonant Vibrational Spectroscopy

NRVS is a nuclear resonant inelastic scattering spectroscopy that measures the vibrational modes (or in other words the created or annihilated phonons) associated with the nuclear resonant transition [1–9]. As illustrated in Figure 1a, while an extremely monochromic incident X-ray beam (e.g., in case of $^{57}$Fe, $E_1$ = 14.41425 keV or 14.4 keV for a simpler description hereafter) scans through the interested nuclear and vibrational transitions, the probed scattering energy is precisely "defined" by the nuclear back radiation at

$E_2 = h\nu_1$ which should have the same linewidth as that for the incoming X-ray beam. Meanwhile, both the nuclear fluorescence at $h\nu_1$ and the converted K-shell electron fluorescence at $h\nu_2$ are collected as the raw counts. These pure nuclear events (lifetime = nanoseconds) can be distinguished and extracted from the often huge (e.g., 20 million cts/s) electronic scattering background (lifetime = femtoseconds) in the time domain [8,10]. Such a time domain-filtered nuclear signal vs. the vibrational energy ($E_{vib} = E_1 - E_2$) forms a raw NRVS spectrum. NRVS becomes available due to the advancement in modern synchrotron radiation (SR) rings, which can provide high intensity/low emittance incoming beams with appropriate time structure(s), advanced high-resolution monochromators (HRM) which reduce the energy bandwidth to 1 meV or narrower (e.g., 0.8 meV at SPring-8 BL35XU, BL19LXU), and fast detectors which distinguish the nuclear events from the huge electronic scattering background. In turn, it presents some distinct advantages in comparison with traditional vibrational spectroscopies such as FTIR [8], resonant Raman spectroscopy [8], and laser-induced fluorescence spectroscopy [11–14].

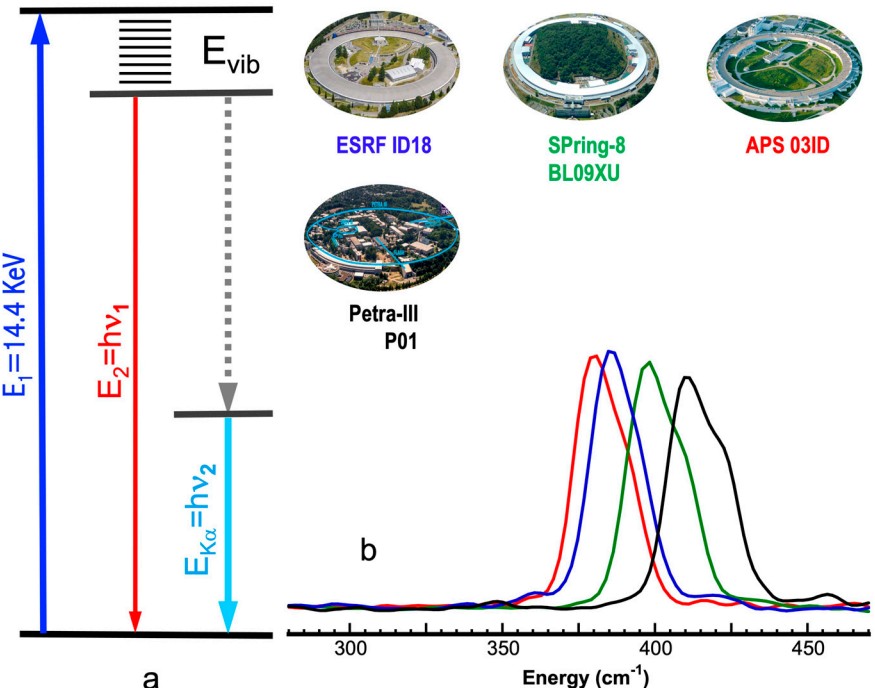

**Figure 1.** (**a**) An illustrative energy diagram for NRVS transitions; (**b**) the uncalibrated NRVS spectra for (NEt$_4$)[FeCl$_4$] measured at four NRS beamlines around the world: APS 03ID (red), ESRF ID18 (blue); SPring-8 BL09XU (green), and Petra–III P01 (black). The top inserts in (**b**) are the overviews for the four SR centers.

Most extraordinarily, NRVS is isotope-specific (site-specific) and is, therefore, an excellent tool to study, for example, Fe-S/P/Cl/O, Fe-CO/CN/NO, and Fe-H/D vibrations [15–18], etc., inside complicated systems in physics, earth sciences, materials sciences, coordination chemistry, and bioinorganic chemistry [10,16–26]. In particular, when a specific site can be selectively labeled with [57]Fe while all the other sites are not labeled [8,27,28], NRVS becomes a pinpoint tool to target the site interested in complicated systems. For example, [2Fe]$_H$ site inside several [FeFe] hydrogenases can be selectively enriched with [57]Fe and all other iron sites are left unenriched [27,28], making all the NRVS signals originate from the [2Fe]$_H$ site. With the same principle, the Fe(II) vs. Fe(III) sites inside a Prussian blue complex or analogs can be alternatively enriched with [57]Fe and probed with NRVS [8,29]. Sometimes different elements can be enriched with different isotopes and measured with different NRVS, e.g., (Et$_4$N)$_3$[[57]Fe$_4$[125]Te$_4$(SPh)$_4$] can be probed with both [57]Fe and [125]Te NRVS [30]. On the other hand, NRVS also has coverage for almost all the

[57]Fe-related vibrations except for a few of them whose [57]Fe atoms do not move during the vibrations [16]. In comparison, either IR spectroscopy or Raman spectroscopy has a "restrictive" selection rule.

Due to the nature of nuclear events, which have much longer decaying times (in the order of ns) than electronic scattering (in the order of fs), and due to the fact that nuclear backscattering has a narrow transition bandwidth, NRVS does not use a diffraction spectrometer and thus has a much better photon in and photon out efficiency [6,8,10,20,22]. Due to the same nature, NRVS also has an almost zero background (after filtering out the electronic scattering background), which leads to the observation of some extremely weak signals, such as the 0.1 cts/s Ni−H−Fe [10,22] or X−Fe−H [27,28,31] features inside various hydrogenase samples. NRVS spectra can also start from a real $0$ cm$^{-1}$ while IR or Raman measurement, on the other hand, often starts from 80–100 cm$^{-1}$.

Due to the simplicity of the NRVS intensity rule (proportional to the displacement of the [57]Fe site in a particular vibrational mode), the pure molecule-based/experiment-independent partial vibrational density of state (PVDOS) can be obtained from all raw NRVS spectra. Therefore, NRVS peak positions, as well as its intensities, can be well reproduced via various theoretical calculations, including density function theory (DFT) [27,28]. In contrast, IR or Raman calculation often involves assumptions about molecular properties such as trail dipole moments or polarizabilities in order to obtain an approximate molecular VDOS.

With regard to practical aspects, NRVS has some compelling practical advantages over established methods. For example, it is water transparent in comparison with far IR spectroscopy, and thus well suited for investigating biological samples in their natural aqueous environment; it is free of fluorescence problems in comparison with resonance Raman spectroscopy, and thus suitable for evaluating photosensitive states; it distinguishes well among O, N, and C in comparison with extended X-ray absorption fine structure (EXAFS) or crystallography [32,33]. Due to these advantages, its user community, as well as the research results, grow exponentially [10,16–25,27,28,31,34]. For example, in the last decade, it has become the third modern X-ray spectroscopy to become popular among biochemical researchers, following crystallography and EXAFS [8,33].

The simplest NRVS scan is an even time scan which measures each data point with the same amount of data acquisition time (also called scanning time), for example five seconds per point (s/p), we call it an even time scan or even scan. For investigating biological samples or other dilute samples, an extremely long scanning time per point must be used in order to obtain spectra with satisfactory statistics. For example, an 89 KDa NiFe hydrogenase molecule has one Fe inside the interested NiFe center, leading to a wt./wt. concentration of ~0.06% Fe in the NiFe center. This does not include the $H_2O$ amount used as the solvent. To probe such a low concentration of Fe, the experimentalists often start to use three seconds per point (s/p) for the low energy region (e.g., from −240 cm$^{-1}$ meV to +400 cm$^{-1}$) to measure Fe−S-related vibrations and 10 s/p for the higher energy region (e.g., +400 cm$^{-1}$~+650 cm$^{-1}$) to measure weak Fe-CO/CN and extremely weak Ni−H−Fe vibrations [22,23]. We call it a sectional scan. When the NRVS features in the Fe−S and Fe−CO/CN regions are well resolved, the experimentalists start to focus on the extremely weak Ni−H−Fe features with 30 s/p scanning time in the corresponding region (e.g., +650 cm$^{-1}$~+800 cm$^{-1}$) while leaving 1 s/p or sometimes even 0.5 s/p for the lower energy region (−240 cm$^{-1}$~+650 cm$^{-1}$). In some extreme cases, 30 s/p will be used for the X−Fe−H features while 1 s/p is used for the nuclear resonant peak region, and the region between the two will be skipped—we call this a jump scan.

## 1.2. Energy Calibration in NRVS

Energy calibration is a procedure that uses a correlation function to map the observed raw energies ($E_{obs}$) onto the calibrated real energies ($E_{real}$). All spectroscopies need energy calibration, those using SR as light sources are no exceptions [8,35–38]. For NRVS, while the high-energy resolution beam from an HRM provides the benefit to resolve detailed

vibrational spectral features, the extremely high resolving power ($>10^7$), which is defined as the exit energy (e.g., 14.4 keV) over its narrow bandwidth (e.g., 0.8 meV), also makes it harder to track and maintain accurate/precise exit energies from an HRM. In other words, it demands more careful energy calibration. For example, in an extreme case, a 0.1 meV energy shift between two spectra can be observed as long as the following two conditions are met: (1) the two spectra in comparison must both have good enough signal-to-noise ratios (S/N) to define and compare the peak centroids with a 0.1 meV difference, and (2) the energy calibration must have a better than 0.1 meV precision/accuracy. While S/N can be improved with more data acquisition time, a good enough energy calibration becomes the key to observing small energy shifts. Although a high energy resolution in the incident beam is also preferable to obtain sharp spectral peaks, this is not a requirement for observing a small energy shift.

According to classical concepts [39], the instrumental scale (ruler) can be calibrated while additional dispersions (instabilities due to environmental differences) are attributed to random/un-calibratable source(s). However, not all the dispersions are imprecisions and at least some energy instabilities are also trackable and "calibratable" [40]. Therefore, there should be two categories of energy calibration for a monochromator, spectrometers, or other instruments. The first one is often performed by the instrument designers or beamline scientists to calibrate a particular device as finely as possible and as repeatably as possible — the aim is to evaluate the design and/or establish the scale for the device. We call this the absolute calibration. To perform an as-fine-as-possible calibration, a high order or more complicated function form is often used to calibrate $E_{obs}$ to $E_{real}$ [41]. For being as repeatable as possible, the calibrations should be performed under a standardized condition, e.g., at 25 K, etc., when the measured correlation between $E_{obs}$ and $E_{real}$ will be stable and repeatable. Although important, this method is not the focus of this publication.

The second is the one that spectroscopists use to calibrate their measured data under whatever the experimental conditions and/or at whatever moment of the measurements—we call this a practical calibration. Such a calibration often omits the higher order corrections and calibrates the energy axis with a linear correlation:

$$E_{real} = (\alpha E_{obs} + E^*) \tag{1}$$

where $E_{real}$, $E_{obs}$, $E^*$ stand for the calibrated real energy, the uncalibrated observed energy, the correction for the zero energy position, respectively, and $\alpha$ is the energy axis scaling factor which needs to measure a standard calibration spectrum [5,6,20] to calculate. All the calibrations discussed in this publication are practical calibrations.

Figure 1b summarizes examples of a series of the uncalibrated spectra for $[FeCl_4]^-$ measured at four nuclear resonant scattering (NRS) beamlines around the world. All these spectra have different energy scales [6,8]—using $\alpha = 0.999$ for APS 03ID, 0.986 for ESRF ID18, 0.965 for SPring-8 BL09XU, and 0.920 for Petra-III P01, all the uncalibrated spectra will be aligned to the published data for its IR spectrum [16]—no figures. If each beamline can have a fixed $\alpha$ value as above, the corresponding $\alpha$ value is referred to as the energy calibration scale for that particular beamline or that particular HRM. Unfortunately, as mentioned in an earlier publication [40], and as will be detailed in the following, this is not the case: the measured $\alpha$ values are different from beamtime to beamtime and sometimes even from different calibration measurements within one beamtime. For example, the measured $\alpha$ values for SPring-8 BL09XU were once between 0.938 and 0.976 [5]. Later, the dispersion span becomes narrower: 0.952–0.966. Still, there is $>0.01$ in its variation span (or $>1\%$ vs. the average $\alpha$ value). BL19LXU is a non-dedicated NRS beamline and its energy scales range from 0.918 to 0.937, dispersing over 2%. For most other NRS beamlines, such variations also exist.

Since most of the Fe−X stretching and bending vibrations have 2–4% in their isotopic shifts e.g., X = $^{13}C/^{12}C$ or $^{36}S/^{32}S$ [5] and even a larger amount in their redox shifts [5,17,23], the current once per beamtime calibration or quick switch calibration [5] still serves the general purposes for NRVS measurements.

It seems no surprise that the measured $\alpha$ values can be different at different moments because practical calibrations were performed under various experimental conditions (e.g., a minor difference in the temperature surrounding the monochromator's crystals). A realistic practice to better track the energy variation is to make more frequent calibrations during NRVS beamtimes. In addition to the conventional energy calibrations which can take six hours in total time [5,6,23], a quick switch calibration procedure was also explored and published earlier [5], in which the incident X-ray beam can be altered between to pass through and measure the main NRVS sample inside the cryostat and to pass right over the main sample's top surface and measure the calibration sample at an room temperature (RT) stage behind the cryostat [5]. This lets calibration measurements be done without moving the main NRVS samples out of the cryostat base, saving the time for sample changing and cryostat temperature cycling. Limited progress has been reported using this calibration: $\leq 0.4\%$ difference was found between two adjacent calibrations when it is performed in every six (6) hours—using tons of beamtime. Nevertheless, there is no guarantee that the scans between the two calibration measurements will have a fixed $\alpha$ value or a predictable dispersion span in its possible values.

### 1.3. Using In-Situ Energy Calibrations

The $\alpha$ values measured with practical energy calibrations should include the real instrument-related scale ($\alpha_{ins}$, which could be equal or at least close to the value from an absolute calibration) and the time-dependent portion ($\alpha_{var}$): $\alpha = \alpha_{ins} \cdot \alpha_{var}$. The conventional NRVS calibration procedures are equivalent to treating the energy instabilities from time to time ($\alpha_{var}$) as un-calibratable imprecision and using whatever the measured or averaged $\alpha$ value to represent the real energy calibration scale for each beamtime. This issue must be resolved because it limits the accuracy of the energy calibration and the overall benefit of NRVS. For measuring and distinguishing small energy shift(s), the time-dependent portion ($\alpha_{var}$) must be tracked, and an in-situ energy calibration procedure becomes necessary. For examples, the 2.4 cm$^{-1}$ isotopic shift ($383.1 \rightarrow 385.5$ cm$^{-1}$) was observed for $[^{57}Fe_4S_4Cl_4]^= \rightarrow [^{X}Fe_4S_4Cl_4]^=$ [5] where $^{X}Fe$ stands for the mixture of $^{57}Fe$ or $^{54}Fe$. In a different case, a 2–4 cm$^{-1}$ peak shift was found in the X$-$Fe$-$H features between the natural (CH) and $^{13}CD$ substituted [FeFe] hydrogenase [31], etc. However, in the latter, when the main NRVS sample has a low concentration in the interested iron site [27,28,31], the experimentalists must leave most of the incident X-ray beam for the main NRVS sample and thus have not much to share for the calibration sample, which often leads to quite noisy calibration spectra [40]. This shows that, although useful in some cases, a direct in-situ measurement is still not an excellent solution to obtain a better or more reliable calibration in most cases.

Another option is to use an environment-related parameter, e.g., temperatures, to track the real exit energies in situ. For example, the four-bounce HRM at APS 03ID uses the angle positions as well as the temperatures at its four crystals to calculate the real beam energies. As a result, the exit beam energies at APS 03ID almost always have an excellent scaling factor around $\alpha = 1$, e.g., 0.999 per the authors' experience as shown in Figure 1b (red curve). However, most other NRS beamlines do not have a temperature monitoring system. The experimentalists must then look for other parameters to represent the temperature variations. According to a recent publication [40], the zero energy drift ($\Delta E_i$, i = scan number index) between adjacent scans becomes the candidate to present and calibrate the variation in the energy scale ($\alpha_i$) although the individual $E_0$ peak positions can just be "used" to correct for the E* value in the conventional calibration procedure [6,8,20]. Instead, the real energies $E_{real}$ can be calculated from the observed energies $E_{abs}$ via the following linear function vs. the accumulated scanning time ($\Sigma t_k$) [40]:

$$E_{real} = [E_{obs} - (\Sigma t_k / T_{tot}) \cdot (\Delta E_i)] \cdot \alpha_0 \qquad (2)$$

where $E_{obs}$, $E_{real}$, and $\Delta E_i$ are defined as above, while the rest variables/parameters are defined as: (1) k = the data point index inside one particular scan while i = the scan number

index; (2) $t_k$ = the data acquisition time at each data point, $\Sigma t_k$ = the accumulated time scanned from the beginning of the scan to the particular point $(1 \rightarrow k)$; and (3) $T_{tot}$ = the total scanning time for the whole scan, $\Sigma t_k / T_{tot}$ is the ratio of the accumulated time scanned (at the point k) to the total time for the whole scan [40]. The Formula (2) can be understood as a stepwise process: (1) $E_{obs}$ is corrected for the energy variation from scan to scan using $E' = [E_{obs} - (\Sigma t_k / T_{tot}) \cdot (\Delta E_i)]$ to create an intermediate energy axis (E'); (2) E' is further converted to the calibrated real energies $E_{real}$ via an additional universal scaling factor $(\alpha_0)$ which should be a constant for each beamline or each HRM [40] (or similar to the $\alpha_{ins}$ as mentioned above). Such a stepwise calibration leads to a much better energy correction than the ones using (1) [40].

In addition to tracking the $\alpha_i$ variation from scan to scan via $\Delta E_i$, Formula (2) also includes the nature that $E_{obs}$ should be mapped onto $E_{real}$ according to the accumulated time scanned at each point. In other words, the energy drift amount per scan ($\Delta E_i$) needs to be distributed to the individual data point (k) according to the ratio of the accumulated time scanned vs. the total time for one scan ($\Sigma t_k / T_{tot}$) rather than scaled with a universal factor [using the Equation (1)]. Although this formula was first proposed in an earlier publication [40], the nature and the application of its time-scaled energy distribution were not elaborated. The focus of that publication [40] was to track and correct the energy variations between different scans rather than within one scan. In this publication, instead, we will focus on: (1) evaluating the distributed energies ($E_{obs} \rightarrow E_{real}$) with the time-scaled function within one NRVS scan or an average NRVS scan while assuming one representing $\Delta E$ value for all NRVS data; (2): analyzing the errorbar contribution due to the improper "distribution" of $\Delta E$ within one scan vs. that due to the different $\Delta E_i$ from different scans. It will be well illustrated later that the former part is as important as or sometimes even more important than the latter part; and (3) establishing a procedure to re-calibrate previous NRVS spectra in case of need. In addition, the case of jump scans where an "unimportant" energy region is skipped will also be evaluated.

Finally, this article aimed to establish the concept that energy positions in one sectional NRVS scan should be tracked and corrected according to the time scanned [as in Equation (2)] rather than be scaled with a universal constant [as in Equation (1)] even though all the NRVS scans have the same zero energy drift value $\Delta E$.

## 2. Experimental Aspects

The NRVS data cited or tested in this study were previously measured at SPring-8 BL09XU or BL19LXU. The science for these NRVS data has been published elsewhere in various scientific journals [10,17,18,22,23,25,27,28]. The raw NRVS data were converted into PVDOS via a PHOENIX software package [42,43] or a PHOENIX inclusive web tool at http://spectra.tools [19,20], which was also detailed elsewhere. As we focused on the energy calibration process, the details about the samples/NRVS measurements/PVDOS conversion are omitted here.

The calibrated value for the zero energy position should always be zero [$E_{real(0)} = 0$]. However, the observed zero energy position [$E_{obs(0)}$ or $E_0$ hereafter] often has a continuous drift in the subsequent NRVS scans (see Figure S1 in the supporting information). The amount of the $E_0$ drift per scan $\Delta E_i$ is defined as $\Delta E_i = E_{0(i)} - E_{0(i-1)}$, where $E_{0(i)}$ is the $E_0$ value for the current scan and $E_{0(i-1)}$ is the $E_0$ value for the previous scan. For the first order approximation, the energy drift amount for the first scan ($\Delta E_1$) is assumed to be the same as that for the second scan ($\Delta E_2$). For previous processed PVDOS spectra, original $\Delta E_i$ values can be further averaged to one $\Delta E$ value to represent the energy drift for the converted PVDOS.

Using $\Delta E_i$ to re-distribute energy drift per scan point by point inside one NRVS scan or using an averaged $\Delta E$ to re-calibrate the previous "calibrated" PVDOS is the central topic of this publication and will be discussed in detail in the following sections.

## 3. Results and Discussions

### 3.1. Understanding Time-Scaled Calibrations

To understand the mechanism for Formula (2) [40], let us start from the basics—the law of crystal diffraction:

$$E_{obs} = hc/(d_0 \cdot \sin\theta_{obs}) \tag{3}$$

$$E_{real} = hc/[d(T) \cdot \sin\theta_{obs}] \tag{4}$$

where T and d(T) are the temperature and the corresponding atomic space of the diffraction crystals inside the HRM at the moment of measurement; $d_0$ is the atomic space at a standard condition (e.g., at 25 K); $E_{obs}$ is the observed energy corresponding to the real energy $E_{real}$; h and c are constants. The comparison of (3) vs. (4) leads to:

$$E_{obs} = [d(T)/d_0] \cdot E_{real} \propto d(T) \tag{5}$$

Therefore, corresponding to one real energy (e.g., at the nuclear resonant peak $E_{real}$ = 14.414 keV), the observed energy $E_{obs}$ is proportional to d(T), or $E_{obs} \propto d(T)$. In short, the energy observed at the HRM ($E_{obs}$) is a function of the monochromator crystals' temperature (T), which is mentioned in the introduction and illustrated here as Equation (5). Although Equation (2) is developed from a pure experimental observation [40], a simplified model of single crystal diffraction and their Equations (3)–(5) provide the readers a theoretical outline of the meaning behind it.

From a qualitative perspective, the trend can be understood in this sequence: once the beam shines on the HRM → the heat load at its crystals increases↑ → crystals' temperature (T) increases ↑ → the crystals' atomic space [d(T)]↑ → $E_{obs}$↑—the longer the time scanned ($\Sigma t_k$), the higher the crystal temperature (T), the larger the energy drift amount ($\Delta E_i$). From a more quantitative perspective, Equation (5) tells us $E_{obs} \propto d$ while the thermal expansion properties for silicon around RT indicates $d \propto T$ [44], therefore $E_{obs}$, d, and T have a linear relationship with one another (or more particularly $E_{obs} \propto T$) within a small variation of temperature T. Since the temperature changing rate is proportional to the temperature difference [e.g., $dT/dt \propto (T_M - T)$] [45,46], its integration shows T (thus d or $E_{obs}$) has an increasing form of the exponential decay function with time t as below [40]:

$$T = T_M - (T_M - T_0) \cdot e^{-kt} \tag{6}$$

$$d = d_M - (d_M - d_0) \cdot e^{-kt} \tag{7}$$

$$E_{obs} = E_M - (E_M - E_{00}) \cdot e^{-kt} \tag{8}$$

where t is the beam-on time from the beginning of the measurement, T is the crystal temperature, d is the atomic spacing of the crystal [=d(T) in (4) and (5)], $E_{obs}$ is the observed energy at the HRM; $T_0$, $d_0$, $E_{00}$ are their initial values at RT; and $T_M$, $d_M$, $E_M$ are their equilibrium limits when the beam is on for a long time and the k here is the Boltzmann constant. The function (8) shows that the observed energy position is the exponential function of the beam-on time. For a first order approach, these exponential functions [especially (8)] can be approximated with a linear function vs. time (t) when we approximate the exponential function with the first two terms of its Taylor series expansion. This explains the nature that the energy drift amount within one scan ($\Delta E_i$) should be "re-distributed" according to a (linear) function of the time scanned rather than a (linear) function of the energy position.

In a practical aspect, Formula (2) can be understood as a stepwise process that includes: (1) $E_{obs} \to E'$ which re-distributes $\Delta E_i$ within each scan and (2) $E' \to E_{real}$ via an additional universal scale with an instrumental-related $\alpha_0$. Since the measured zero energy point [such as E* in (1)] can always be calibrated in the final data analysis in PHOENIX or so [5,8], we can make any initial alignment between the observed and calibrated energies—it is not necessary to align them to their nuclear resonant peak, as in Formula (1). The NRVS measured at SPring-8 scans in the direction from the higher energy end (let's call it E2')

to the lower energy end (E1′), it is easier to align the observed $E2_{obs}$ and the intermediate energies E2′, i.e., set $E2_{obs}$ = E2′, as illustrated in Figure 2a vs. Figure 2b. To illustrate the concept, we assumed an exaggerated larges ΔE which is half of the size of (E2′ − E1′)—that is $(E_{obs}2 - E_{obs}1) = (\frac{1}{2}) \cdot (E2' - E1')$ in Figure 8. When the scan reaches the ending energy at the left end (the setpoint at E1′), its observed energy position $E1_{obs}$ should have drifted $+\Delta E_i$ in comparison with its target (E1′) or, in another word, E1′ should have a $-\Delta E_i$ correction in comparison with the observed $E1_{obs}$: E1′ = $E1_{obs} - \Delta E_i$, as shown in Figure 2b vs. a. A positive energy drift ($\Delta E_i > 0$) means the E1′ is lower in energy position than $E1_{obs}$, and vice versa. Here, we in fact have assumed that the amount of energy drift when HRM scans from E2′ to E1′ is the same as the $E_0$ position drift ($\Delta E_i$) between the current and the previous scans, which is almost true.

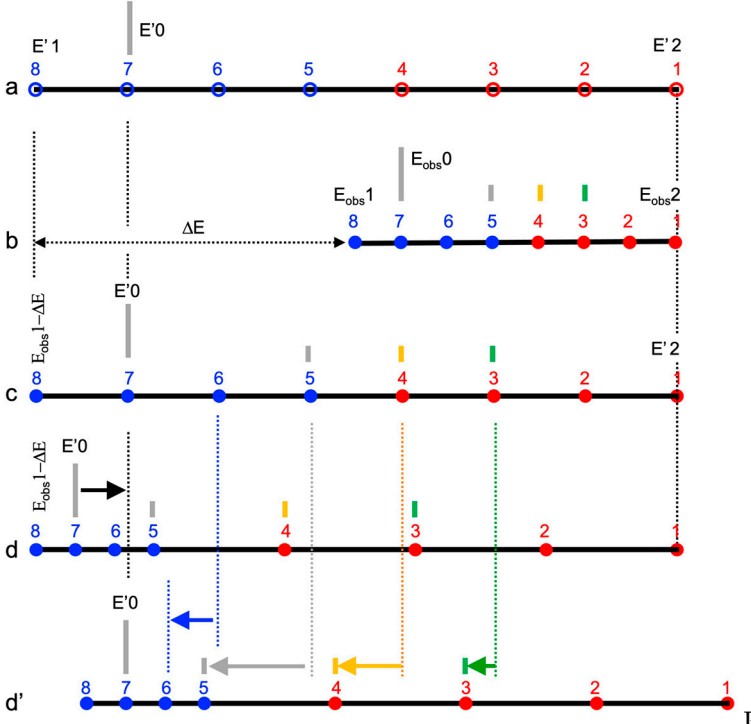

**Figure 2.** A conceptual NRVS measurement which scans from the higher energy end (E2′) to the lower energy end (E1′) and which scans points 1, 2, 3, 4 (red) with 3t s/p and points 5, 6, 7, 8 (blue) with t s/p: (**a**) a targeted theoretical range for the NRVS spectrum (in E′); (**b**) the observed raw energies (at points 1−8) when we assume an exaggerated larges ΔE which is half of (E2′ − E1′)— that is $(E_{obs}2 - E_{obs}1) = (\frac{1}{2}) \cdot (E2' - E1')$; (**c**) the corrected intermediate energies (E′) which map the large amount ΔE onto the 8 data points using universal energy scale Formula (1); (**d**) the corrected intermediate energies (E′) which map the large amount ΔE onto the 8 data points using time-scaled correlation (2); and (**d′**) the corrected energies in (**d**) with the $E_0$ position aligned to 0.

When the scan is arriving at a middle point between E2′ and E1′, the energy drift between the observed $E_{obs}$ and the ΔE corrected E′ accumulates gradually and continuously, starting from 0 to $\Delta E = (\frac{1}{2}) \cdot (E2' - E1')$. According to (8), this partial amount should be in general following an exponential decay function vs. the beam-on time (t). Within a short period (e.g., one hour or one scan), however, a linear function can almost approximate the relationship between the energy amount drifted (at point k) vs. the beam-on time t from point 1 to point k ($\Sigma t_k$). Therefore, the $E_{obs}$ energy position at one particular point k should have drifted a part of the—ΔE (or $\Delta E_i$ if each NRVS scan is processed individually)

that is proportional to the beam-on time which is the Equation (2). When an even scan is performed, the result from the Equation (2) is the same as that from the Equation (9) [40]:

$$
\begin{aligned}
E' &= E_{obs} - \{(E2_{obs} - E_{obs})/(E2_{obs} - E1_{obs})\} \cdot \Delta E_i \\
&= E_{obs} \cdot \{1 + \Delta E_i/(E2_{obs} - E1_{obs})\} - \{E2_{obs} \cdot \Delta E_i/(E2_{obs} - E1_{obs})\}
\end{aligned}
\tag{9}
$$

Under this condition, it is reasonable to uniformly scale the $E_{obs}$ according to the Equation (9), rather than transform it point by point with the Equation (2). Therefore, it is the nature of a sectional scan that makes it necessary to correct the energies point to point according to the accumulated beam-on time ($\Sigma t_k$), which is (2).

### 3.2. Comparing Two Calibration Procedures

Figure 2 assumes an exaggerated 8-data-point NRVS file that scans from the higher energy end $E2'$ (right) to the lower energy end $E1'$ (left), mimicking the real situation for the data measured at SPring-8. Figure 2a illustrates the theoretical energy range in $E'$ in mind when a scan is planned while (b) illustrates the observed raw energies ($E_{obs}$). The data points 1, 2, 3, 4 (red) are measured with 3t s/p, and the points 5, 6, 7, 8 (blue) are measured with t s/p, 1/3 of that for the points 1, 2, 3, 4. This conceptual spectrum has a zero vibrational energy position at point 7 and a few assumed "NRVS peaks" at points 5 (grey), 4 (yellow), 3 (green). We also assume an exaggerated but illustrative energy drift amount of $\Delta E$ which is equal to half of the targeted energy span ($E2'$–$E1'$). When such an NRVS is scanned, we can first align the starting points for the observed and intermediately calibrated NRVS and let $E2_{obs} = E2'$. When the NRVS reaches its ending points, its observed energy should have an energy drift of $+\Delta E$ in comparison with the targeted energy $E1'$ (or $E1' = E_{obs}1 - \Delta E$). This $E1'$ should have the same energy value no matter whether we use the calibration procedure (9) [equivalent to (1)] or the procedure (2) to calibrate it, such as shown in Figure 2 vs. d: Figure 2c illustrates the $E'$ which is calibrated according to a universal energy scale (9) and 2d illustrates the $E'$ which is calibrated according to the accumulated beam-on time [according to (2)]. The re-distributed $E'$ for the two calibrations in the middle region between $E1'$ and $E2'$ are very different: Figure 2c vs. d.

However, the peak position difference between Figure 2c and d is not the real difference between the energy positions obtained from the two calibration methods. To compare, the zero vibrational energy positions for 2c and 2d should first be aligned with each other, e.g., aligned to 0, shifting 2d to 2d'. Comparing Figure 2c vs. d', the peak position difference is the real difference between the two energy calibration methods. It is obvious that the peak position difference is relatively small near $E_0$ or near the start of the scan ~$E2'$. On the other hand, the energy difference for the peaks around the changing point for the data acquisition time, e.g., at points 4 and 5, is much larger than for the other peaks. The NRVS experimentalists often care more about the peaks above the time changing point because that is often the region of interest.

Figure 3 illustrates another imaginary NRVS file but with more realistic scan parameters including an assumed total energy drift of 0.8 meV per scan in a total scanning span of 1000 cm$^{-1}$. Figure 3a is an example of scanning time (s/p) at each data point: the time is not to the scale but the scanning time per point is noted on the grey horizontal line: 30 s/p is used from 800 to 650 cm$^{-1}$ while 1 s/p is used for the rest (low energy) regions. Although Figure 3 is from pure conceptual data, such a 30 s/p vs. 1 s/p scanning time has often been used to scan the Ni–H–Fe wagging mode in *DvM*F [NiFe] hydrogenase [10,22] and the X–Fe–H bending features for several [FeFe] hydrogenases [28,31] in a similar region. Figure 3b shows the energy re-distribution of the assumed 0.8 meV energy drift within one NRVS scan: the red curve represents the re-distribution of this 0.8 meV according to the accumulated time scanned [using the Equation (2)]; the blue curve shows the even spread of the 0.8 meV via a procedure similar to the Equation (9). Although the energies are in general different for the two energy correction methods, the two curves in (b) (blue and red) begin at the same starting point ($E2' = E2_{obs}$) and reach the same ending point at $E1' = E1_{obs} - 0.8$ meV. The difference between the two calibration methods (blue vs. red)

reaches its maximum at 650 cm$^{-1}$ when the scanning time per second changes from 30 s/p to 1 s/p, which is consistent with the conclusion from Figure 2.

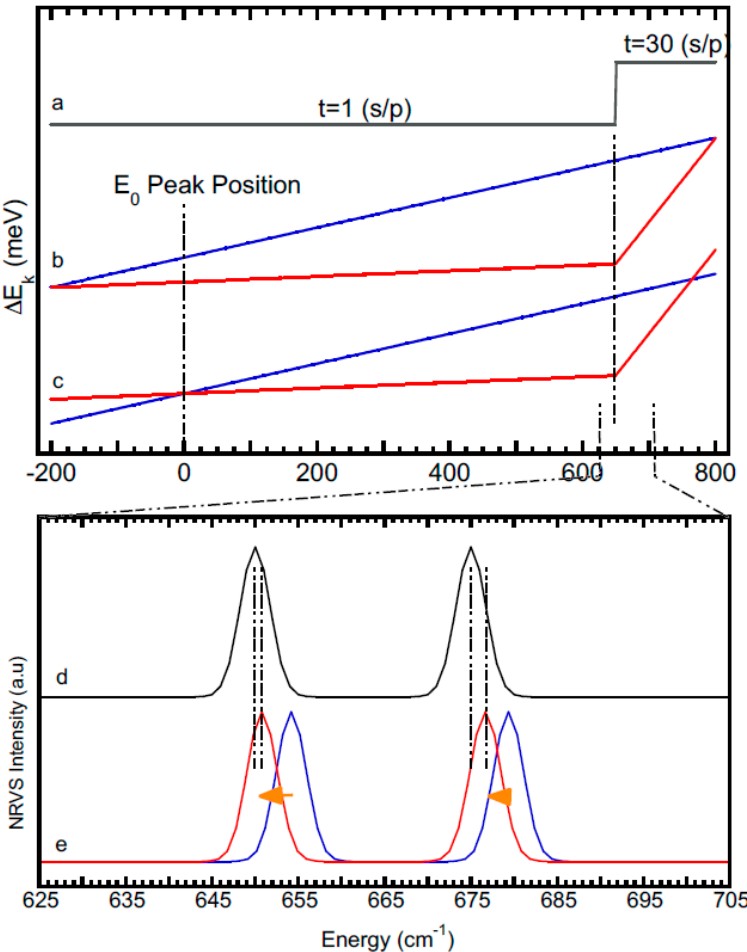

**Figure 3.** (**a**) An example scanning time per data point (s/p) used in a conceptual NRVS sectional scan between −200 cm$^{-1}$ and +800 cm$^{-1}$: the scanning time is not to the scale but it is noted on the grey horizontal line; (**b**) the energy shift per data point assuming a total 0.8 meV energy drift per scan: the energy drift at each point calibrated with Equation (9) (blue) vs. that calibrated according to the accumulated time scanned with (2) (red); (**c**) the results from (**b**) with the zero vibrational peak position moving to 0 cm$^{-1}$ in both calibration methods; (**d**) a conceptual NRVS spectral section between 625 and 705 cm$^{-1}$; and (**e**) the NRVS spectra calibrated with the two calibration methods used in (**c**) (color match).

In addition to the analysis with formulas, the two calibration processes can also be understood figuratively as the following. The region of 650–800 cm$^{-1}$ accounts for (150/1000) = 15% of the whole energy range (from −200 cm$^{-1}$ to 800 cm$^{-1}$) but takes $30*150/(30*150 + 1*850) = 84\%$ of the total scanning time. Therefore, an energy-scaled traditional calibration procedure [the Equation (9)] leads to 15% of the −0.8 meV (or −6.45 cm$^{-1}$) being assigned to this region while a time-scaled calculation procedure [the Equation (2)] leads to 84% of the −0.8 meV (or −6.45 cm$^{-1}$) being assigned to the same region, leaving −4.6 cm$^{-1}$ difference between the two calibration methods at 650 cm$^{-1}$, as shown in Figure 3b. However, this −4.6 cm$^{-1}$ is not the real difference between the two calibration methods. These curves must still align their zero vibrational energies with each other which lead to the curves in Figure 3c. After the zero energies for the two calibrations are re-aligned to 0 cm$^{-1}$ as shown in Figure 3c, the maximum difference at 650 cm$^{-1}$ becomes about 3.7 cm$^{-1}$ instead, which is the real difference between the two "re-distribution" methods.

Figure 3d illustrates one assumed NRVS spectrum in the region of 625–705 cm$^{-1}$ with two peaks at 650 and 675 cm$^{-1}$ respectively. The calibrated spectra using the above two calibration methods are shown as in Figure 3e [blue = the spectrum calibrated with a universal energy scale using the Equation (9) vs. red = the spectrum calibrated with time-based re-distribution using the Equation (2)]. Although both methods (e, blue or red) lead to larger peak position shifts in comparison with the original un-calibrated data (d, black) at the higher energy peak (675 cm$^{-1}$), it is the peak at 650 cm$^{-1}$ which has the larger difference (error) between the two different calibration methods (blue vs. red). The value of this error varies according to the scanning details of the sectional scans. It is 0.46 meV (or 3.7 cm$^{-1}$) here.

As mentioned in the introduction [40], the Equation (2) was initially used to account for the $\alpha_i$ variation due to the different $\Delta E_i$ values from different NRVS scans [40]. Then, the errorbar due to the difference between different scans is estimated as ($\Delta E_i - \Delta E_{cal}$) = 0.8 − 0.3 = 0.5 meV (=4.03 cm$^{-1}$), where $\Delta E_i$ corresponds to the energy drift for different scans while $\Delta E_{cal}$ stands for the energy drift under which the calibration scan is measured (which is assumed at 0.3 meV here). In comparison, under an assumed 0.8 meV energy drift per scan and the above scan parameters, the $E_{err}1$ (0.46 meV) is almost the same as $E_{err}2$ (0.5 meV). In case that a $\Delta E_i$ = 0.6 meV is assumed, the $E_{err}1$ will be 0.35 meV while $E_{err}2$ = 0.6 − 0.3 = 0.3 meV: $E_{err}1 > E_{err}2$. Therefore, $E_{err}1$ is a critical portion of the total errorbar that needs to be resolved.

### 3.3. Re-Calibrating Published PVDOS

According to the above discussion, when the energy drift amount per scan ($\Delta E_i$) has about the same level, the errorbar due to the energy variation from scan to scan ($E_{err}2$) can be omitted. However, for sectional scans, the "internal" re-distribution of the averaged energy drift ($\Delta E$) still causes an obvious (if not significant) errorbar ($E_{err}1$). How can we "re-calibrate" the old NRVS to include the time-scaled re-distribution using the Equation (2)?

The first task is to "reverse the original calibration process" using the original energy scale constant $\alpha$: $E^*_{obs} = E^*_{real}/\alpha$, where $E^*_{real}$ is the energy that was already "calibrated" with the traditional calibration procedure (1). In this step, we need to use the original value of $\alpha$ from the old analysis record. Most published NRVS spectra were already in the form of PVDOS which was averaged and transferred from a series of raw NRVS spectra [6,42,43]. Such spectra (PVDOS in $E^*_{real}$) often have $E^*_{real(0)}$ = 0 while the $E^*_{obs}$ can in principle have any value including 0, so we do not need to care about it here. In case of need, it will be automatically shifted in the final analysis with PHOENIX [5,43], just like the constant c in an infinity integration.

The second step is to convert the pseudo "raw" data just obtained (in $E^*_{obs}$) to that in a new intermediate energy scale $E'$ using the first portion of the Equation (2). For processing individual scans, $\Delta E_i$ is needed to convert it from $E^*_{obs}$ to $E'$. For processing an averaged PVDOS, a uniform $\Delta E$ is needed, which can be obtained by averaging the $\Delta E_i$ per scan from the original raw data corresponding to the PVDOS.

The last step is to convert the data in $E'$ to that in $E_{real}$ using $E_{real} = E' \cdot \alpha_0$ which needs a scaling constant $\alpha_0$. This constant is in principle a beamline-dependent constant and must be obtained via a large amount of energy calibration data using a previously proposed stepwise procedure [40]. At the moment, only the $\alpha_0$ value for SPring-8 BL19LXU has been evaluated as 0.918—a preliminary number [40]. Before the $\alpha_0$ values become available for all other beamlines, they can be estimated by using the Equation (1) [or its alternative form (9)] and (2):

$$\alpha_0 = \alpha/\{1 + \Delta E/(E2^*_{obs} - E1^*_{obs})\} \tag{10}$$

which shows that $\alpha_0$ can be estimated from $\alpha$ (for the original energy calibration), $\Delta E$ (which is averaged from the $\Delta E_i$ for the original raw data), and ($E2_{obs} - E1_{obs}$) (which is the energy range scanned). A combination of all the above three steps leads to:

$$E_{real} = [E^*_{obs} - (\Sigma t_k/T_{tot}) \cdot (\Delta E)] \cdot \alpha_0$$
$$= [E^*_{real}/\alpha - (\Sigma t_k/T_{tot}) \cdot (\Delta E)] \cdot \alpha/\{1 + \Delta E/(E2^*_{obs} - E1^*_{obs})\} \quad (11)$$

where $E^*_{real}$ is the energy axis from the previously "calibrated" NRVS (PVDOS) while $E_{real}$ is the newly calibrated energy axis processed with the new time-scaled calibration procedure (2), all other variables are defined as before.

When a published PVDOS needs to be re-calibrated, in addition to the published PVDOS (in $E^*_{real}$), the original NRVS data are also needed to find out the $t_k$ and $\Sigma t_k$ as well as the $\Delta E_i$ to calculate an averaged energy drift $\Delta E$. The $\alpha_0$ also must be calculated from $\alpha$ value that was used in the previous calibration according to the Equation (10).

### 3.4. Examples of Re-Calibrated NRVS

Let us start to look at a few real cases. [FeFe] hydrogenases are complex metalloenzymes, which can lead to hydrogen production in numerous organisms [47], and therefore have potential importance in the post-carbon era economy. The mechanism of $H^+$ transfer, $Fe-H$ or $Fe-H_2$ bonding, and $H_2$ bond activation are all associated with the aminodithiolate (ADT) group inside the $[2Fe]_H$ sub-cluster which is the enzyme's active center. Chemically, scientists have successfully developed artificial maturation [26–28] which can replace the original NH at the ADT with an O atom to form an oxodithiolate (ODT) variant or to change the functional group at the surrounding position 169 from –SH to –OH (from Cystine to Serine), in addition to enriching a few selected iron sites with $^{57}$Fe while leaving other irons unenriched. Spectroscopically, the site-specific NRVS can extract just the $^{57}$Fe-related vibrations and thus is better to distinguish the minor differences among different enzymatic variants. For example, two $X-Fe-H$ wagging and bending vibrational peaks are observed from *Chlamydomonas reinhardtii* [FeFe] hydrogenase (or *Cr* HydA1 for short) and its variants [27,28], with one at ~675 cm$^{-1}$ and another in the region of 725–775 cm$^{-1}$. The first peaks are almost at the same location for all the variants but the positions for the second peaks are different corresponding to the different variants: ODT variant (green) → wild type enzyme (blue) → C169S variant (grey), as illustrated in Figure 4a. The detailed science and meaningfulness are as published [27,28] and will not be repeated here. In short, understanding the variation of these $X-Fe-H$ peaks and their shifts corresponding to the variations in and around the NH site is relevant to the catalytic mechanism of the hydrogenases.

As these peaks are in the energy region where the 30 s/p data acquisition time is used while the front region is scanned with a 1 s/p data acquisition time, the peak positions can have an obvious error in the traditional energy calibration (1) as discussed above. This can be re-corrected with a time-scaled energy re-distribution process (2) or (11). With $\Delta E$ and $\alpha$ which are obtained from the original data set and analysis record, Equation (11) produces a new PVDOS with re-distributed energies ($E_{real}$) shown as a red solid curve in Figure 4b. For a direct comparison, the traditionally calibrated PVDOS for the same sample is also duplicated from Figure 4a to Figure 4b as a dashed blue line. A small shift of ~3 cm$^{-1}$ is observed [Figure 4b blue vs. red]. Although shifts are not great, it is large enough to affect the comparison of some small energy shifts, such as the 2 cm$^{-1}$ as mentioned in the introduction for the CH-ADT and its C$^{13}$D substituted counterpart [31]. We also noticed that the peak at 675 cm$^{-1}$ has a bit more energy difference between the two calibration methods than the peak at 750 cm$^{-1}$ does, consistent with the analysis in Figures 2 and 3.

In addition to the $X-Fe-H$ bending features, NRVS for the $Fe-H$ stretching mode is also observed but at the current time only for model complexes. Since the mass ratio of H:Fe is 1:57, most of the motion in the $Fe-H$ stretch mode is at H, not at Fe. This nature leads to a very weak NRVS signal for the $Fe-H$ feature although it is still the best method to observe a $Fe-H$ vibration. For example, several $Fe-H$ and $Fe-H_2$ stretching modes in *trans*-[$^{57}Fe(\eta^2$-$H_2)(H)(dppe)_2$][$BPh_4$] ($H_2FeH$ for a simple description) were published [48] and are re-illustrated as in Figure 5a, again with the traditional calibration process (1) (blue,

corresponding to E*$_{real}$). The vibrational modes associated with the NRVS peaks at 1773 and 1915 cm$^{-1}$ were assigned to the asymmetric Fe–H stretching mode from the Fe(H$_2$) component and the Fe–H stretching from the "local" FeH bond [48]. The peak at 1956 For cm$^{-1}$ is the Fe–H stretching mode from a byproduct when H$_2$ accidentally dissociates from the complex [48]. Nevertheless, this peak is the NRVS feature with the highest energy observed to date. When we use the original α value, the averaged energy drift per scan ΔE, and Equation (11) to re-calculate the energy transformation E*$_{real}$ → E$_{real}$, the obtained spectrum is illustrated in Figure 5a (red) while the original spectrum is shown n the same figure as blue. To better illustrate the details, Figure 5b,c show the same spectra in two narrow regions.

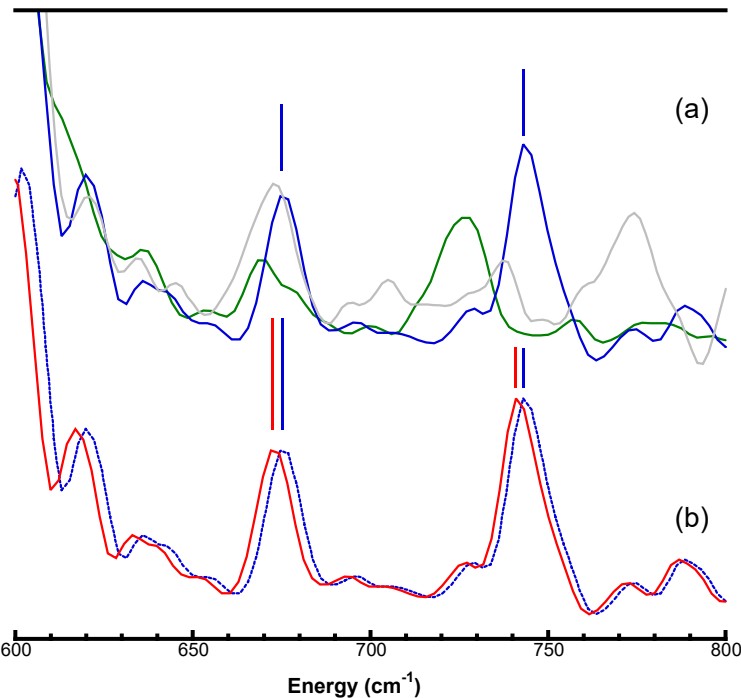

**Figure 4.** (**a**) NRVS PVDOS in the X–Fe–H region for *Cr* [FeFe] hydrogenase in the H$_{hyd}$ form calibrated with the traditional procedure (1): green = ODT, blue = wild type, grey = C169S; and (**b**) NRVS PVDOS for the sample spectrum of wild type enzyme re-calibrated with the time-scaled procedure (2) (red) vs. that calibrated with the traditional procedure (1) (dashed blue): the duplication of the wild type spectrum in 4a.

In this particular case, 20 s/p was used for 1600–2000 cm$^{-1}$ while 0.1 s/p was used for the lower energy region. It is clear that the closer the peak position is to the scanning time changing point, the higher the energy shift amount is. We observed that there is a −3.4 cm$^{-1}$ for the peak at 1773 cm$^{-1}$; a −2.3 cm$^{-1}$ for the peak at 1915 cm$^{-1}$; and a −1.8 cm$^{-1}$ for the peak at 1756 cm$^{-1}$.

In Figure 2, the peak position lower than the timing changing position also has obvious differences between the two calibration procedures, such as point 5 in Figure 2c vs. Figure 2d′. In a real NRVS, for example, the Fe−CO peaks in the *DvM*F NiFe hydrogenase can have obvious differences in their peak positions in the spectrum calibrated with the two different calibration procedures. This is well illustrated in Figure 6. On this side, the lower the peak position, the smaller the energy difference between the two calibration procedures: e.g., −3.0 cm$^{-1}$ for the Fe−CO peak at 606 cm$^{-1}$; −2.4 cm$^{-1}$ for the Fe−CO peak at 548 cm$^{-1}$. People usually do not care much about the lower energy side because the region(s) are usually scanned in much less time than the focused higher energy region. Nevertheless, in cases where the Fe−CO peaks are to be used as internal calibration marks

for tracking the X−Fe−H bending peak positions, the issues discussed here become critical to consider.

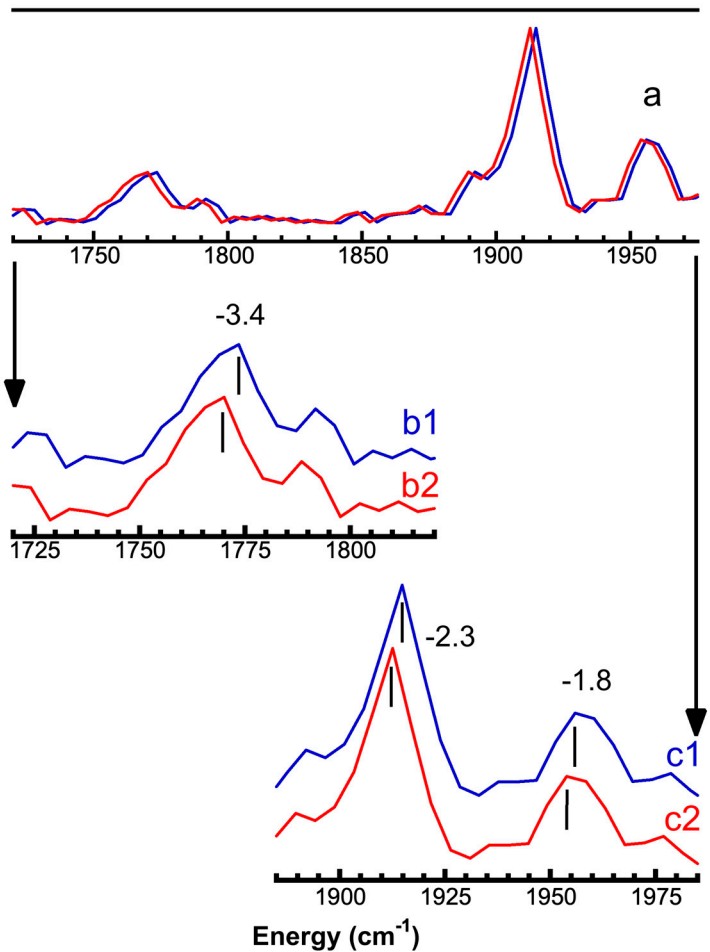

**Figure 5.** (**a**) NRVS PVDOS of complex $trans-[^{57}Fe(\eta^2-H_2)(H)(dppe)_2][BPh_4]$ ($H_2FeH$ for short) in the region for Fe–H stretching modes calibrated with the traditional procedure (1) (blue) and with new time-scaled redistribution method (2) (red); (**b**) PVDOS for (**a**) in the narrower region of $1720-1820$ cm$^{-1}$; and (**c**) PVDOS for (**a**) in the narrower region of 1885–1985 cm$^{-1}$.

### 3.5. Dealing with Jump Scans

A sectional scan does not necessarily include just two sections; multiple sections or other formats of time structure are also possible. In one example, 30 s/p can be used for the interested region, such as the above X−Fe−H features [27,28], while 1–2 s/p can be used for the nuclear resonant peak region. The region between these two scan regions can be skipped without scanning—we call this type of scan a jump scan. Such a jump scan is often used to measure: (1) the NRVS whose general feature is well observed and only a small portion of interested region needs to be measured further, such as the X−Fe−H features [27,28]; (2) a particular feature in unenriched samples, such as or the Fe−CN for the unenriched $Fe(CN)_6$ related complexes which is the only measurable peak [29,49–51].

Figure 7a first exhibits the NRVS for $^{57}Fe$ enriched $(NH_4)_2Mg(II)[^{57}Fe(II)(CN)_6]$ [49] [29] as a reference. It has a prominent feature around 594 cm$^{-1}$, which corresponds to $Fe(II)(CN)_6$ core vibrations [29]. Other features are much weaker. For evaluating unenriched (similar) complexes, the weak features are even weaker and are probably out of the detection limit [the 2% $^{57}Fe$ in natural abundance vs. the 100% $^{57}Fe$ in the fully enriched samples] [29]. For those cases, measuring a full NRVS scan is usually out of question and probing the prominent feature at 594 cm$^{-1}$ becomes the only hope. Therefore, a jump scan will be useful. For example, measured NRVS for unenriched $KEu(III)Fe(II)(CN)_6$ is shown as in Figure 7b. It is

calibrated with the traditional procedure (1) (blue), which scales energies with a universal factor (blue) and the new calibration procedure (2) (red), which re-distributes the energies according to the time-scanned ($\Sigma t_k$). Figure 7c illustrates the seconds used to scan each data point—the text clearly shows 2 s/p for the nuclear resonant peak and 50 s/p for the Fe-CN region. As each scan took about 1.5 h (relatively long), and the beam is not very stable at the moment of the measurement, the energy drifts (the $\Delta E_i$ values) are ~1.5 meV per scan (a bit higher than the regular scans"). Therefore, the re-calibrated Fe-CN peak has about a $-6$ cm$^{-1}$ change towards the lower energy direction.

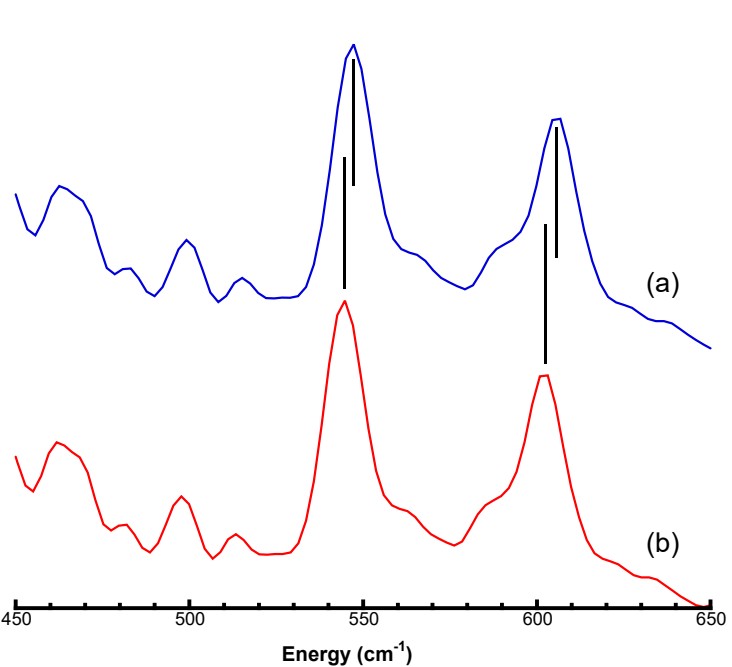

**Figure 6.** The FeCO feature [in *DvMF* NiFe hydrogenase NRVS] in the lower energy side of the 30 s/p to 1 s/p timing changing point: (**a**) the NRVS spectrum calibrated with the traditional calibration procedure; and (**b**) the NRVS spectrum re-calibrated with the time-scaled calibration procedure (2).

### 3.6. Further Discussions

Figure 8 summarizes a few conceptual cases with different scan parameters but the same energy drift per scan (e.g., $\Delta E$ = 0.8 meV). The calibrated energy value for each data point ($E_{real(k)}$) is calculated using Equation (2) and distributed according to different scanning parameters. For simpler consideration, we only consider the first calibration step, $E_{obs} \to E'$ in the following discussions (Figure 8). The $\Delta E_k$ is defined as the energy correction from the $E_{obs}$ to $E'$ at one particular point (k). For different calibration procedures, the $\Delta E_k$ should have different values at one particular point (k). However, their starting energy ($E2' = E2_{obs}$) and ending energy ($E1' = E1_{obs} - \Delta E$) are always the same no matter how the energies between the two ends are calibrated. The blue curves in Figure 8a,b illustrate the cases with an even time NRVS scan as the reference to discuss other scans. Figure 8a starts with an extreme case: a "jump" scan with 30 s/p scanning time for $800 \to 650$ cm$^{-1}$ and 0 s/p for the rest $650 \to -200$ cm$^{-1}$ (red). The difference between this scan (red) and an even time scan (blue) is clear with the largest difference occurring at $650$ cm$^{-1}$ where the scanning time changes from 30 s/p to 0 s/p Two additional examples of "jump" scans are also provided in Figure 8a where the scanning time changes from 30 s/p to 0 s/p at 400 and 0 cm$^{-1}$ respectively (the 2 black curves). As a general rule, the following "conclusions" hold:

(1)   it is obvious that the wider the scanning region (vs. the skipped region) [$800 \to 650$ cm$^{-1}$) $\to$ ($800 \to 400$ cm$^{-1}$) $\to$ ($800 \to 0$ cm$^{-1}$)], the closer its energy distribution curve to

the case of an even NRVS scan. For example, the left black curve is the closest one to the blue curve in Figure 8a;

(2) the time ratio of the two adjacent scanning regions dictates the difference between the energies calculated via the Equation (1) and those calculated via the Equation (2): for example, the 5:10 [Figure 8b, light blue] provides a much closer result to the even time scan (blue) than the 1:30 (purple) or 30:0 (red) does. This leads to the concept that a multiple section scan that changes the scanning time at a gradual pace is closer to the even time scan. In practical measurements, it is better to start with an even time scan or a sectional scan but with multiple and stepwise changes in its scanning time parameters. When the major features are all well resolved and calibrated, some extremely weak features need to be probed with heavy counting on one region, such as the 30/1 or the 30/0 s/p scans discussed in Figure 8a. It becomes necessary to use the time-based energy calibration procedure (2) for any sectional scans but it is especially necessary for the ones with large time steps, such as the 30/1 or 30/0 ones;

(3) no matter which calibration procedure is to be used and no matter what the scanning parameters, no point should have an energy drift amount greater than the total energy drift per scan ($\Delta E_i$). Therefore, the final factor for controlling the possible energy calibration errors is to scan NRVS with a lesser $\Delta E_i$ value. For example, NRVS experimentalists need to avoid the moments right after each hutch opening or so and wait for the beam to be stabilized. For further reduction of $\Delta E_i$ values, NRVS users also have the option to use less time for each scan and take more scans to average.

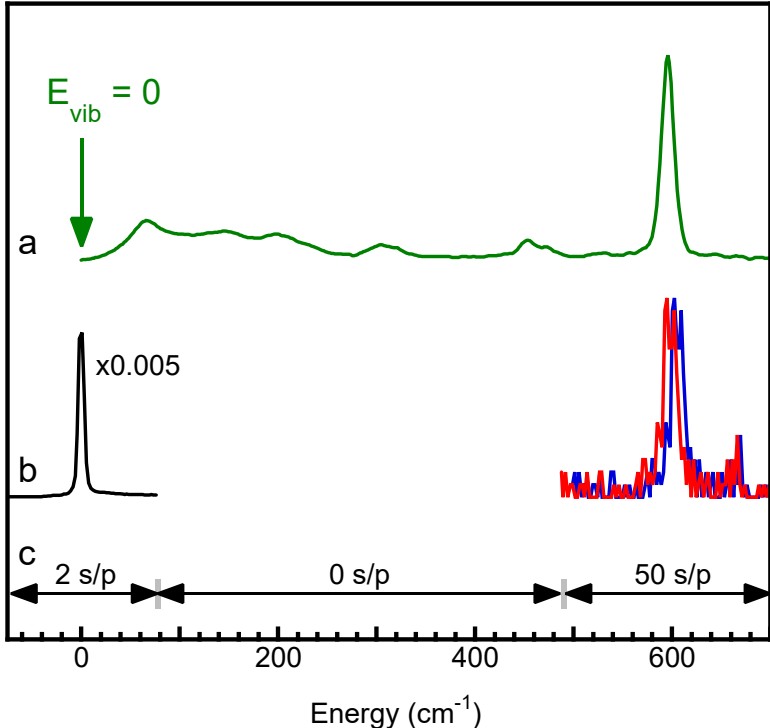

**Figure 7.** The NRVS spectra for a series of $Fe(CN)_6$ containing complexes: (**a**) PVDOS for $(NH_4)_2MgFe(CN)_6$; (**b**) raw NRVS spectrum for $KEuFe(CN)_6$ which was measured with a jump scan procedure around the Fe(CN) region and the nuclear resonant peak region. The blue spectrum was calibrated with the tradition procedure (1) and the red one is re-calibrated with a time-based procedure using (2); and (**c**) the scanning time per point (s/p) used in different energy regions during the NRVS measurement for $KEuFe(CN)_6$ (**b**).

Although the examples used in this publication are the data from [57]Fe NRVS, NRVS for other isotopes (e.g., [125]Te [30], [40]K [52], [151]Eu [29,53]) should have the same principle

discussed here—energy should be calibrated according to the time scanned rather than the energy position travel because the energy drifts according to the beam-on time.

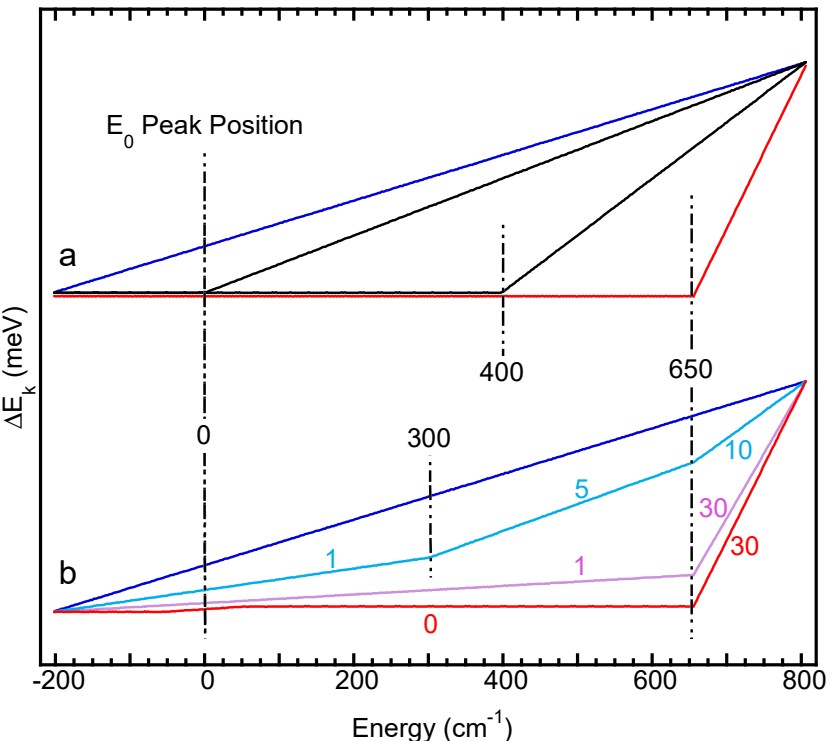

**Figure 8.** The energy shift per data point ($\Delta E_k$) distributed from an assumed total 0.8 meV energy drift per scan using the Equation (2) but for NRVS data with different scanning parameters: (**a**) an even time scan (blue) vs. the 30/0 s/p jump scan (red) which changes the scanning time at 650 (red), 400 (right black) and 0 (left black) cm$^{-1}$ respectively; and (**b**) an even time scan (blue) vs. the 10/5/1 s/p sectional scan (light blue), the 30/1 s/p sectional scan (purple) and the 30/0 jump scan (red).

## 4. Summary

In this publication, we evaluated the energy calibration ($E_{obs} \rightarrow E_{real}$) with the time-scaled function within one NRVS scan according to the Equation (2). The errorbar contributed from the improper "distribution" of $\Delta E_i$ within one scan (or $\Delta E$ for the averaged scan) ($E_{err}1$) vs. that due to the different $\Delta E_i$ (and thus different $\alpha_i$) from scan to scan ($E_{err}2$) were analyzed and compared. It was found that the former ($E_{err}1$) is as important as or sometimes even more important than the latter ($E_{err}2$).

In case of need, a procedure [the Equation (11)] was established to re-calibrate previous NRVS spectra (PVDOS) with $E^*_{real}$ as their energy axis to new spectra with $E_{real}$ as their energy axis. For practical purposes, one averaged $\Delta E$ value calculated from the old raw NRVS data is used to re-calibrate the old NRVS-derived PVDOS. Sectional scans with different scanning conditions as well as jump scans where an "unimportant" energy region is skipped from the scanning are also discussed in detail.

In the end, via this article, the concept can be established that energy positions in one sectional NRVS scan should be corrected or re-distributed according to the time scanned [as in the Equations (2) or (11)] rather than be scaled with a universal constant [as in the Equation (1)].

Although the Formula (2) itself was mentioned in an earlier publication [40], it is this publication that elaborated on the meaningfulness and the applications of the energy re-distribution within one NRVS scan. Several published NRVS spectra were re-calibrated

with a few cm$^{-1}$ energy shifts in comparison with the published spectra (PVDOS) calibrated with a traditional energy calibration (1). These spectra are shown as in Figures 4–7.

The transition from the old calibration procedure (1) to the new stepwise calibration practice takes time and some users may prefer to continue using the old calibration procedure. Even without tracking the energy variation from scan to scan due to different $\Delta E_i$ {Wang, 2022 #55}, using an averaged $\Delta E$ for all the NRVS scans and redistributing this averaged $\Delta E$ according to the time scanned [using the Equation (2)] rather than according to a universal scale [using the Equation (1)] can fix the $E_{err}1$ and reduce the total errorbar by ~50%.

**Supplementary Materials:** The following supporting information can be downloaded at: https://www.mdpi.com/article/10.3390/physchem2040027/s1, Figure S1: A series of NRVS spectra for $^{57}$Fe powder: the $E_0$ position drifts to the higher-energy direction as scanning time increases (red → orange → yellow → green → cyan → dark blue → purple).

**Author Contributions:** H.W. and Y.Y. measured the NRVS data and proposed the initial concept, J.W. streamlined and tested the details in the mathematical modeling. All the authors contribute equally. All authors have read and agreed to the published version of the manuscript.

**Funding:** This research received no external funding.

**Acknowledgments:** We thank Stephen P. Cramer at SETI Institute (USA) for his overall support for this project via US NIH grants GM-65440. The NRVS spectra used, cited, mentioned, or tested for this publication were measured at SPring-8 BL09XU or BL19LXU (e.g., proposals 20200013, 20210033 via RIKEN, and 2018A1409, 2018B1379, 2018A1033, 2019A1259 via JASRI).

**Conflicts of Interest:** The authors declare no conflict of interest.

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
