# Peer review of "The True Nature of the Energy Calibration for Nuclear Resonant Vibrational Spectroscopy: A Time-Based Conversion"

_2673-7167, doi:10.3390/physchem2040027_

Round 1

Reviewer 1 Report

In this work, the authors evaluated the distributed energies with the time scaled function within one NRVS scan. The error bar was analyzed, compared, and calibrated. A procedure is established to re-calibrate previous NRVS spectra. In my opinion, this work is of practical significance and worth publication as is, or after minor revision.

1. Figure 7. could the authors rescale the x and y-axis to make the comparison easier? 

2. Figure 6, please provide the whole spectra with this range zoomed in, instead of the current form.

Author Response

Dear Reviewer,

Thank you very much for your support for our manuscript.

  1. Figure 7. could the authors rescale the x and y-axis to make the comparison easier? 

Thank you very much for your suggestion, I revised the Figure 7 in the revised manuscript.

  1. Figure 6, please provide the whole spectra with this range zoomed in, instead of the current form.

Showing the whole spectra will just make it more crowded. I request to keep the current format. We did show a few examples in other figures (e.g.. Figure 7).

Reviewer 2 Report

This is a very interesting article and in my opinion deserves publication.

I have, nevertheless, a few comments which I believe may improve the article:

1.- In page 5:

"Unfortunately, this is not the case – the measured (alpha) values are different from beamtime to beamtime and sometimes even within one beamtime."

Perhaps you could expand on why ".. and sometimes even within one beamtime"

2.- Page 5, 3rd paragraph

"It seems no surprise that the measured (alpha) values can be different at different moments because practical calibrations were performed under various experimental conditions"

Perhaps you could expand what these "various experimental conditions" are.

3.- Page 6 (2 lines before the end of the 2nd paragraph)

It is mentioned Reference 41, but this reference is invalid!!

4.- Page 6 (last paragraph before Section 2)

It says: "as in (2)" and "as in (1)".  To improve clarity I suggest to write: "as in equation (2)" and "as in equation (1)"

5.- Page 7

First line, it says "It is not perfect but better than nothing", it is far too colloquial.  I suggest the use of rigorous scientific terms in order to describe "how much it is not perfect" and also, what do you exactly mean by "nothing".

6.- Page 7 (just below equation (5))

Again it is mentioned Reference 41, which is non existant!!

7.- Page 7, idem

"Although equation (2) is developed from a pure experimental observation [41], a simplified model..."

Since Reference 41 does not exist you should provide an argument for the validity of equation (5).

8.- Page 7 (just before equation 6)

Makes reference to the "early study" of reference 41, which does not exist.

9.- Page 7, idem

It is mentioned that Eobs, d and T have a linear relationship with one another.  How long it is linear? Can you get into the nonlinear region?

10.- Paragraph below equation 8

You mention "for a first order approach...". What should we expect at the second o higher order?

11.- Idem

"This explains the nature that the energy drift amount within one scan (DeltaEi) should be “re-distributed” according to a linear function of the time scanned"

At 5 times "k" an exponential Exp(-kT) is at zero and Eobs will not have any change.  Therefore the constant "k" define the minimum observation time to obtain a fixed Eobs value.  Could you discuss this?

12.- Page 8. Below equation 9

Your argument is supported by reference 41 which is non-existant.

13.- Page 16. Three lines before section 3.5

"As each scan took about 1.5 hours, and the beam is not super stable at the moment of the measurement, the DeltaEi values are ~1.5 meV".  If were the case where the scan takes 3 hours or 10 hours. How much the situation would improve?

14.- Page 18. Just before the Summary

"Although the above discussion is using the data from 57Fe NRVS measurements, NRVS for other isotopes (e.g. 125Te[29], 40K[52], 151Eu[53]) should have the same principle and similar energy distribution issues as discussed here".  Your argument is based on the validity of an extrapolation. How sound and valid this is?

15.- Page 18

Several times the non-existant reference 41 is mentioned.

Author Response

Dear Reviewer,

0) I am sorry that the [41] is actually the [42]. Also, it is published now, so I changed [41] to [42] and revised the page information for [42] as well. I reformatted the manuscript after that. Thank you very much for your detailed reading and questions, which I believe will certainly make this manuscript better. For the questions, I will answer them as the following.

1.- In page 5:

"Unfortunately, this is not the case – the measured (alpha) values are different from beamtime to beamtime and sometimes even within one beamtime."

Perhaps you could expand on why ".. and sometimes even within one beamtime"

Before “this is not the case,” I added ”as mentioned in an earlier publication and as will be detailed in the following”;

Added “from different calibration measurements” within the sentence to make it clearer.

2.- Page 5, 3rd paragraph

"It seems no surprise that the measured (alpha) values can be different at different moments because practical calibrations were performed under various experimental conditions"

Perhaps you could expand what these "various experimental conditions" are.

Add “(e.g. a miner difference in the temperature surrounding the monochromator’s crystals) ”

3.- Page 6 (2 lines before the end of the 2nd paragraph)

It is mentioned Reference 41, but this reference is invalid!!

Answered at the beginning of the comments to the reviewer 2.

4.- Page 6 (last paragraph before Section 2)

It says: "as in (2)" and "as in (1)".  To improve clarity I suggest to write: "as in equation (2)" and "as in equation (1)"

Added “the equation“ here and other places (all highlighted in yellow or purple).

5.- Page 7

First line, it says "It is not perfect but better than nothing", it is far too colloquial.  I suggest the use of rigorous scientific terms in order to describe "how much it is not perfect" and also, what do you exactly mean by "nothing".

As it seems not necessary, we deleted this sentence in the revised version.

6.- Page 7 (just below equation (5))

Again it is mentioned Reference 41, which is non existant!!

Answered at the beginning of the comments to the reviewer 2.

7.- Page 7, idem

"Although equation (2) is developed from a pure experimental observation [41], a simplified model..."

Since Reference 41 does not exist you should provide an argument for the validity of equation (5).

Answered at the beginning of the comments to the reviewer 2.

8.- Page 7 (just before equation 6)

Makes reference to the "early study" of reference 41, which does not exist.

Answered at the beginning of the comments to the reviewer 2.

9.- Page 7, idem

It is mentioned that Eobs, d and T have a linear relationship with one another.  How long it is linear? Can you get into the nonlinear region?

Revised to: “From a more quantitative perspective, equation (5) tells us Eobs   (proportional to) d while the thermal expansion properties for silicon around RT indicates d (proportional to) T [45], therefore Eobs, d, and T have a linear relationship with one another (or more particularly Eobs  (proportional to) T) within a small variation of temperature T.”

10.- Paragraph below equation 8

You mention "for a first order approach...". What should we expect at the second o higher order?

Linear function is the first term of Taylor expansion, other higher orders will contribute less. Therefore, for the first order approach, we only take the first order. Anyway, linear regression is always the first order approach for the measurement based correlations which we use.

We added “when we approximate the exponential function with the first two terms of its Taylor series expansion” to link exponential function to linear function.

11.- Idem

"This explains the nature that the energy drift amount within one scan (DeltaEi) should be “re-distributed” according to a linear function of the time scanned"

We added: “rather than a (linear) function of the energy position” at the very end of the paragraph for clarity.

At 5 times "k" an exponential Exp(-kT) is at zero and Eobs will not have any change.  Therefore the constant "k" define the minimum observation time to obtain a fixed Eobs value.  Could you discuss this?

Actually, the energy E is a function of time: exp(-kt) or a linear function with approximation. This is already discussed. If you mean something else, please comment again. Thank you very much.

12.- Page 8. Below equation 9

Your argument is supported by reference 41 which is non-existant.

Answered at the beginning of the comments to the reviewer 2.

13.- Page 16. Three lines before section 3.5

"As each scan took about 1.5 hours, and the beam is not super stable at the moment of the measurement, the DeltaEi values are ~1.5 meV".  If it was the case where the scan takes 3 hours or 10 hours. How much the situation would improve?

Actually the longer the scanning time the larger the energy amount drifts. For best results, the users need to take many short scans instead. However, as the energy scan has dead time per point, extremely short scans does not make sense. One hour scan is usually preferred (actually for any SR based spectroscopies). As the discussed sample in Figure 7 is NOT 57Fe labelled, its signal level is very low, we have to use very long scanning time (1.5 hr) to deal with the issue.

14.- Page 18. Just before the Summary

"Although the above discussion is using the data from 57Fe NRVS measurements, NRVS for other isotopes (e.g. 125Te[29], 40K[52], 151Eu[53]) should have the same principle and similar energy distribution issues as discussed here".  Your argument is based on the validity of an extrapolation. How sound and valid this is?

The principle should be the same for sure. However, to avoid confusion, we deleted “and similar energy distribution issues as ” and change it to “should have the same principle discussed here – energy should be calibrated according to the time scanned rather than the energy position travel. ”

15.- Page 18

Several times the non-existant reference 41 is mentioned.

Answered at the beginning of the comments to the reviewer 2.

Reviewer 3 Report

The paper by Wang et al. deals with the proper correction to be performed for energy calibration based on time scanned, in the framework of nuclear resonant vibrational spectroscopy. By means of theoretical, numerical and mainly experimental procedures, the roles of a point-by-point temporal-based conversion or of a constant one with a universal average value are investigated. After reporting the state-of-the-art for NRVS, and examining the energetic drift and scaling factor, the authors study a bunch of data from previous measurements, and present a set of equations starting from basic crystal diffraction. Focusing then on graphical representations all showing inverse centimeters on the axis of abscissae, they analyze in detail sectional and jump scans, and conclude on the best redistribution for minimizing the errorbar. Their results are carefully presented and plotted, and the relevance of these findings with respect to the existing bibliography from scientific literature is thoroughly discussed.

In my opinion, the article deserves publication on Physchem. Hereafter I only mention few minor issues, to be addressed by the authors.

1) At the beginning of the abstract, the acronym NRVS is mistakenly introduced twice.

2) Page 2: "which distinguishes" -> "which distinguish"

3) Page 4: "there should" -> "there should be"

4) Fifth line of page 7: "and using or" -> "or using", "are the" -> "is the"

5) Page 9: "a energy" -> "an energy"

6) Label in figure 3: "Energ" -> "Energy"

7) In the caption of figure 5, the item (c) seems to be missing.

8) Page 14: "here becomes" -> "here become"

9) Page 15: "necessary to" -> "necessarily"

10) Page 16: item 1) of the conclusions should start a new line

11) Page 18: "was analyzed" -> "were analyzed"

12) Page 18: the main verb is missing in the paragraph beginning with "In the end"

13) Reference [41] is incorrect

Author Response

Dear Reviewer,

Thank you very much for your detailed reading and suggestions, which I believe will certainly make the manuscript better and which I will answer them one by one - all the revised locations are highlighted in yellow (unless the text is deleted).

1) At the beginning of the abstract, the acronym NRVS is mistakenly introduced twice.

Yes. I deleted the second one. 

2) Page 2: "which distinguishes" -> "which distinguish"

 Yes. Revised.

3) Page 4: "there should" -> "there should be"

 Revised to “there should be ”

4) Fifth line of page 7: "and using or" -> "or using", "are the" -> "is the"

Revised accordingly.

5) Page 9: "a energy" -> "an energy"

Revised accordingly.

6) Label in figure 3: "Energ" -> "Energy"

Revised accordingly.

7) In the caption of figure 5, the item (c) seems to be missing.

“(c)” is added.

8) Page 14: "here becomes" -> "here become"

Revised accordingly.

9) Page 15: "necessary to" -> "necessarily"

Revised accordingly.

10) Page 16: item 1) of the conclusions should start a new line

Revised accordingly.

11) Page 18: "was analyzed" -> "were analyzed"

 Revised accordingly.

12) Page 18: the main verb is missing in the paragraph beginning with "In the end"

 Add “the concept can be established” at the beginning of this sentence.

13) Reference [41] is incorrect

Replaced with [42] and reformatted in the revised manuscript.

Reviewer 4 Report

The paper from Hongxin Wang and coworkers evaluated the distributed energies within the NRVS scan and present the contribution of errorbar from the ΔEi or ΔE. The paper is well organized and results are well demonstrated. In particular, the background part provides enough information to help reader understand the history and necessity of the research. I believe the work is well suited for Physchem. I only have one question for the research.

1. The research is pretty interesting. The relative intensity of various peaks in spectrum is also important in research. Like the red and black lines in Figure 1. The shoulder can be clearly observed in black like and hard to be noticed in the red line. I would like to question if this can also be included in calibration in the future

Author Response

Dear Reviewer,

  1. The research is pretty interesting. The relative intensity of various peaks in spectrum is also important in research. Like the red and black lines in Figure 1. The shoulder can be clearly observed in black like and hard to be noticed in the red line. I would like to question if this can also be included in calibration in the future.

Thank you very much. Actually, under the current experimental condition, the showing up of this should varies from time to time from location to location. Therefore, we only use the peak centroid to calibrate the energies. We will keep evaluating it in the future.

Round 2

Reviewer 2 Report

Accept